# REINITIALIZING WEIGHTS VS UNITS FOR MAINTAINING PLASTICITY IN NEURAL NETWORKS

## ABSTRACT

Loss of plasticity is a phenomenon where a neural network loses its ability to learn when trained for an extended time on non-stationary data. It is a crucial problem to overcome when designing systems that learn continually. An effective technique for preventing loss of plasticity is reinitializing parts of the network. In this paper, we compare two different reinitialization schemes: reinitializing units vs reinitializing weights. We propose a new algorithm named *selective weight reinitialization* for reinitializing the least useful weights in the network. We compare our algorithm to continual backpropagation, a previously proposed algorithm that reinitializes units. Through our experiments in continual supervised learning problems, we identify two settings when reinitializing weights is more effective at maintaining plasticity than reinitializing units: (1) when the network has a small number of units and (2) when the network includes layer normalization. Conversely, reinitializing weights and units are equally effective at maintaining plasticity when the network is of sufficient size and does not include layer normalization. We found that reinitializing weights maintains plasticity in a wider variety of settings than reinitializing units.

Systems that learn from a continuous stream of data, that *learn continually*, are better suited for making predictions about a changing world such as ours. For example, a system that learns continually in a water treatment plant makes more accurate predictions than a system that learns offline and is then deployed (Janjua et al., 2023). Similarly, a system that continually adjusts its predictions about drivers' earnings makes better ride-sharing matches than alternatives based on fixed heuristics (Azagirre et al., 2024). Even current large language model systems such as ChatGPT (OpenAI, 2023) could be improved if they are designed to learn continually; such systems could stay up to date with current information without needing to be retrained from scratch. The already impressive performance of modern deep learning systems could be further improved if such systems are designed to learn continually.

However, modern deep learning systems were designed using the train-once approach, in which networks are trained once on a large dataset, then frozen and deployed. Unfortunately, the techniques developed under the train-once approach often are unsuccessful in continual learning. A form of failure of conventional deep learning systems is the loss of the ability to learn when the system is trained for an extended time on non-stationary data, a phenomenon known as *loss of plasticity* (Dohare et al., 2024). Since the essential requirement of a learning system is that it is capable of learning from data, loss of plasticity presents a fundamental problem for deep learning systems that learn continually.

Fortunately, loss of plasticity can be prevented. An effective and simple technique for mitigating loss of plasticity is sporadically reinitializing parts of the network. Reinitialization algorithms must carefully balance maintaining plasticity and preserving the information stored in the network weights. If a large part of the network is reinitialized at once, then the information necessary for making correct predictions may be destroyed, harming performance. On the other hand, if reinitialization is done too sparsely, the network might still suffer from loss of plasticity. Continual backpropagation (Dohare et al., 2021; 2024) achieves this balance by occasionally reinitializing the least useful units in the network. To date, continual backpropagation is one of the most effective reinitialization algorithms for maintaining plasticity (Kumar et al., 2024).

The idea of reinitializing parts of the network can be implemented at many different levels, such as the entire network (Nikishin et al., 2022), a number of layers (Nikishin et al., 2022; Dohare et al., 2024), units (Dohare et al., 2021; Sokar et al., 2023; Dohare et al., 2024), or weights in the network. Of all the ways reinitialization can be implemented, reinitialization at the level of the weights has yet to be studied for the purpose of maintaining plasticity. This paper fills this gap in the literature by proposing an algorithm for reinitializing weights in the network named *selective weight reinitialization*. Every certain number of updates, selective weight reinitialization measures the utility of the weights in each layer in the network and reinitializes a proportion of the weights with the lowest utility.

Using selective weight reinitialization and continual backpropagation, we empirically investigate the question: are there settings where reinitializing weights is more effective at maintaining plasticity than reinitializing units? We first study this question with feed-forward networks in the permuted MNIST problem (Goodfellow et al., 2014; Zenke et al., 2017), where we found that reinitializing weights is more effective at maintaining plasticity in two settings: (1) when the network has a small number of units per layer and (2) when the network employs layer normalization. We then proceed to compare both algorithms in a class-incremental learning problem based on the CIFAR-100 dataset (Krizhevsky et al., 2009) using residual networks (He et al., 2016) and vision transformers (Dosovitskiy et al., 2021). Once again, we found that reinitializing units is less effective at maintaining plasticity when the architecture includes layer normalization, such as in vision transformers. However, when combined with another reinitialization scheme, which resets the parameters of the layer normalization modules, reinitializing units is just as effective as reinitializing weights at maintaining plasticity. Overall, we found that reinitializing weights successfully maintained plasticity in a wider variety of settings than reinitializing units, suggesting it is a more reliable reinitialization scheme.

Our study uncovers settings where the well-studied reinitialization scheme, reinitializing units, loses plasticity. We contribute towards a general solution for maintaining plasticity in neural networks by proposing a new reinitialization scheme that reinitializes weights. This new scheme prevents loss of plasticity in settings where reinitializing units fails to maintain plasticity. In addition to maintaining plasticity, reinitializing weights has the added benefit of being straightforward to implement. Measuring the utility of units in a network architecture has to account for the complex connectivity patterns of the different structures in the network. On the other hand, reinitializing weights does not have to account for any complex interdependencies between structures, so it can be readily applied to any network architecture.

## 1 RELATED WORK

### 1.1 LOSS OF PLASTICITY

Recently, the loss of plasticity effect has drawn the attention of the machine learning community. At first, the observations were presented in different subfields in machine learning such as class-incremental learning (Chaudhry et al., 2018), supervised learning (Ash & Adams, 2020), reinforcement learning (Dohare et al., 2021; Nikishin et al., 2022; Lyle et al., 2022), and continual learning (Dohare, 2020; Rahman, 2021), but the observations were not recognized as part of the same underlying phenomenon. However, it was only when Sutton & Dohare (2022) presented a direct study of the phenomenon that the community developed a unifying language, and all previous observations were attributed to the same underlying phenomenon. Since then, an increasing number of papers has studied the loss of plasticity effect in the last couple of years (Abbas et al., 2023; Sokar et al., 2023; Lyle et al., 2023; 2024; Lee et al., 2024b;a; Elsayed & Mahmood, 2024; Elsayed et al., 2024; Dohare et al., 2024; Lewandowski et al., 2024; Kumar et al., 2024).

Along with the direct study of loss of plasticity, several algorithms have been proposed to mitigate the effect. Proposed techniques for maintaining plasticity include regularizing the parameters of the network (Kumar et al., 2024; Lewandowski et al., 2024; Dohare et al., 2024; Elsayed et al., 2024), architectural modifications (Lyle et al., 2023; Abbas et al., 2023; Nikishin et al., 2023; Lyle et al., 2024; Lee et al., 2024b), adding parameter noise (Ash & Adams, 2020; Elsayed & Mahmood, 2024), and, the focus of this paper, reinitialization techniques (Nikishin et al., 2022; Dohare et al., 2021; Sokar et al., 2023; Dohare et al., 2024). Moreover, combining multiple techniques is often more effective at maintaining plasticity than any of them alone (Lee et al., 2024a; Dohare et al., 2024). We contribute to this rich literature by proposing a new reinitialization algorithm that maintains

plasticity in a wide variety of settings. Our algorithm can be easily applied to any architecture and combined with other methods for maintaining plasticity.

## 1.2 REINITIALIZATION ALGORITHMS

Reinitialization algorithms have been employed for maintaining plasticity (Nikishin et al., 2022; Dohare et al., 2021; Sokar et al., 2023; Dohare et al., 2024) and for improving generalization performance in neural networks (Mahmood & Sutton, 2013; Taha et al., 2021; Alabdulmohsin et al., 2021; Zhou et al., 2022; Zaidi et al., 2023). Reinitializing layers in a network has been shown to increase the decision margins and promote convergence to a flatter local minimum, resulting in improved generalization in supervised learning (Alabdulmohsin et al., 2021). For loss of plasticity, reinitialization has been used to restart dormant or dead units in the network and restore the initial conditions of the weights that promote learning (Sokar et al., 2023; Dohare et al., 2024). These algorithms vary in what parts of the network they reinitialize, such as the entire network, groups of layers, single layers, and units.

Reinitialization is also an integral part of dynamic sparse training algorithms. While the primary goal of such algorithms is to directly learn a sparse network, the algorithms often involve pruning and restarting weights in the network (Mocanu et al., 2018; Evci et al., 2020). The motivation of dynamic sparse training algorithms is to explore the space of subnetworks in a larger network to find a sparse solution (Frankle & Carbin, 2019). These algorithms have been shown to be robust to periodic changes in their input distribution, which suggests that they may also be effective for maintaining plasticity (Grooten et al., 2023). The algorithm we introduce in this paper, selective weight reinitialization, has parallels to dynamic sparse training algorithms. However, instead of learning a sparse network, we entirely focus on reinitializing weights to maintain plasticity.

Finally, there is a biological basis for reinitialization algorithms. Biological neurons have been observed to prune a proportion of their synaptic connections periodically along with growing new connections at the same rate (Kasai et al., 2021). This process is analogous to the continuous initialization of weights in reinitialization algorithms. The synaptic pruning and growing process in biological neurons suggest that reinitialization may be a requirement to facilitate continual learning in connectionist networks. Notably, reinitialization happens at the level of synaptic connections, equivalent to weights in neural networks, not at the level of neurons.

## 2 LEARNING PROBLEM

We focus our study of plasticity to the continual supervised learning setting. In this setting, a learning system generates predictions, $\hat{\boldsymbol{y}} \in \mathbb{R}^c$, based on observations, $\boldsymbol{x} \in \mathbb{R}^n$, to match a target, $\boldsymbol{y} \in \mathbb{R}^c$. Observations and targets are jointly sampled from a probability distribution $p$, which changes every certain number of samples, $S$. For convenience, we refer to all the observation-target pairs sampled from the same probability distribution as a task. We subscript the probability distribution of each task by $k$. Thus, observation-target pairs are sampled according to $p_0$ in the first task, $p_1$ in the second task, and so on. On each task, the goal of the learning system is to minimize the expected loss between its predictions and the targets $\mathbb{E}_{p_k}[\ell(\boldsymbol{y}, \hat{\boldsymbol{y}})]$. For the rest of the paper, we use the cross-entropy loss $\ell(\boldsymbol{y}, \hat{\boldsymbol{y}}) = -\sum_{j=1}^c y_j \log(\hat{y}_j)$.

We use a neural network parameterized by $\boldsymbol{\theta}$ to generate predictions in the continual supervised learning setting, $f_{\boldsymbol{\theta}}(\boldsymbol{x}) = \hat{\boldsymbol{y}}$. At learning step $t \in \{0, 1, \ldots, S-1\}$ in a task, the network receives a mini-batch of $m$ observation-target pairs, $\{(\boldsymbol{x}_i, \boldsymbol{y}_i)\}_{i=1}^m$, sampled from the probability distribution of the current task, $p_k$ for $k \geq 0$. To keep track of the evolution of the learning system, we subscript the parameters of the network by the current learning step and the current task number, $\boldsymbol{\theta}_{S \cdot k+t}$. Since access to $p_k$ is not often available, the network parameters are updated to minimize the empirical loss $\hat{J}(\boldsymbol{\theta}_{S \cdot k+t}) = \frac{1}{m} \sum_{i=1}^m \ell(\boldsymbol{y}_i, f_{\boldsymbol{\theta}_{S \cdot k+t}}(\boldsymbol{x}_i))$ based on the current mini-batch of data. To update the network parameters, we use the stochastic gradient descent rule,

$$\boldsymbol{\theta}_{S \cdot k+t+1} \doteq \boldsymbol{\theta}_{S \cdot k+t} - \alpha \nabla_{\boldsymbol{\theta}_{S \cdot k+t}} \hat{J}(\boldsymbol{\theta}_{S \cdot k+t}),$$

where $\alpha \geq 0$ is a learning rate parameter that scales the size of the update and $\nabla_{\boldsymbol{\theta}_{S \cdot k+t}} \hat{J}(\boldsymbol{\theta}_{S \cdot k+t})$ is the gradient of the empirical loss with respect to the current network parameters.

To measure loss of plasticity, we compare the learning performance in the current task of a network trained continually on all previous tasks and a newly initialized network. If the performance of the network trained continually is lower than the performance of the newly initialized network, then the network trained continually has lost plasticity. In the permuted MNIST experiments in Sections 3 and 4, we use the accuracy computed as the network is learning as a measure of performance, *online accuracy*. In the incremental CIFAR-100 experiments in Section 5, we use the accuracy computed on a separate test set, *test accuracy*. Both of these metrics measure the ability of a network to generalize to unseen data, which is different from the ability to minimize the loss studied in other papers (Lyle et al., 2023; Elsayed & Mahmood, 2024). Henceforth, we use loss of the ability to generalize and loss of plasticity interchangeably. Still, we note that loss of plasticity has been used to refer to both loss of trainability (Lyle et al., 2023; Lewandowski et al., 2024) and loss of generalizability (Ash & Adams, 2020; Lee et al., 2024b; Dohare et al., 2024).

## 3 REINITIALIZING WEIGHTS FOR MAINTAINING PLASTICITY

Several reinitialization schemes have been used for the purpose of maintaining plasticity. However, one remains to be explored in the loss of plasticity literature: reinitializing weights. The first contribution of this paper is to propose an algorithm that reinitializes weights and to study its effectiveness at maintaining plasticity.

We named our algorithm *selective weight reinitialization*. Every certain number of updates, selective weight reinitialization measures the utility of the weights in the network and reinitializes a proportion of the weights with the lowest utility. The motivation for reinitializing parts of the network is to restore the initial conditions that allowed the network to learn and that were slowly removed by the learning process. The algorithm involves four different design choices: the utility function, $U$, used for ranking the weights, the reinitialization strategy, $\mathcal{R}$, which dictates how to reinitialize parameters in the network, the reinitialization frequency, $\tau$, and the proportion of weights, $p$, to be reinitialized at each reinitialization step.

We study two utility functions: a utility function based on the magnitude of the weights, *magnitude utility*, and a utility function based on the magnitude of the gradient of the weights, *gradient utility*. Given a weight, $w$, in a matrix, $\boldsymbol{W}$, the magnitude utility function assigns a utility of $|w|$ to the weight. The gradient utility function assigns a utility of $|w \cdot g_w|$, where $g_w$ is the derivative of the loss with respect to $w$, which can be estimated from a mini-batch of data. Both utility functions are widely used in neural network pruning with comparable results (Blalock et al., 2020), and both can be implemented with little computational overhead.

We devise two reinitialization strategies based on the initialization distribution used at the start of training. The first reinitialization strategy samples new values from the initialization distribution. For example, if the entries of a matrix, $\boldsymbol{W}$, were initialized according to a Normal distribution with mean $\mu$ and standard deviation $\sigma$, then when reinitializing $w \in \boldsymbol{W}$, we sample its new value from $\mathcal{N}(\mu, \sigma)$. On the other hand, if the entries of the bias vector, $\boldsymbol{b}$, were initialized to a fixed value of zero, then at reinitialization $b \in \boldsymbol{b}$ would be set to zero. We call this reinitialization strategy *reinitialization with initial distribution*. We chose this reinitialization strategy because it moves the distribution of weights closer to the initialization distribution, which is designed to facilitate learning (Glorot & Bengio, 2010; He et al., 2015). The second reinitialization strategy reinitializes weights to the mean of their initialization distribution. Using the same example as before, $w$ would be reinitialized to $\mu$, and $b$ would be reinitialized to zero. We call this reinitialization strategy *reinitialization to the mean*. Since the mean of initialization functions is often zero, this strategy is equivalent to setting the value of new weights to zero. Setting the values of new connections to zero yields good generalization performance in dynamic sparse training algorithms (Mocanu et al., 2018; Evci et al., 2020).

Note that there would be a reinitialization function for each weight matrix and bias vector in the network for either of these reinitialization strategies. We represent a reinitialization strategy as a set of reinitialization functions with an entry for each weight matrix and bias vector. Thus, for a network parameterized by $\{\boldsymbol{W}_1, \boldsymbol{W}_2, \dots\}$, omitting bias vectors for simplicity, the corresponding reinitialization strategy is $\mathcal{R} = \{I_1, I_2, \dots\}$, where $I_i$ is the reinitialization function for a matrix $\boldsymbol{W}_i$. Given a utility function and a reinitialization strategy, the reinitialization frequency, $\tau$, and reinitialization proportion, $p$, are treated as hyper-parameter values to be tuned. Finally, when computing the number of weights to reinitialize, we handle decimal numbers by sampling from a Bernoulli distribution

---

**Algorithm 1** Selective Weight Reinitialization

---

**Input:** network with $L$ hidden layers with weights $\{\boldsymbol{W}_1, \ldots, \boldsymbol{W}_L\}$
**Input:** utility function $U$
**Input:** reinitialization strategy $\mathcal{R} = \{I_1, \ldots, I_L\}$
**Hyper-parameters:** Reinitialization frequency $\tau$ and reinitialization proportion $p$
**for** each training step $t$ **do**
    Sample a mini-batch of data
    Compute prediction, loss, and gradients, and update network parameters
    **if** $t$ is a multiple of $\tau$ **then**
        **for** $\boldsymbol{W}_i$ in $\{\boldsymbol{W}_1, \ldots, \boldsymbol{W}_L\}$ **do**
            Compute utilities: $\{U(w) \mid w \in \boldsymbol{W}_i\}$
            Compute number of weights to reinitialize:
            $k \leftarrow \text{Integer Part}(p \cdot |\boldsymbol{W}_i|) + Bernoulli(\text{Fractional Part}(p \cdot |\boldsymbol{W}_i|))$
            Reinitialize the value of the $k$ lowest-utility weights using $I_i$
        **end for**
    **end if**
**end for**

---

with a probability of success equal to the decimal number. Algorithm 1 gives the pseudocode for selective weight reinitialization.

We proceed to assess the effectiveness of selective weight reinitialization at maintaining plasticity. We assess the four combinations of utility functions and reinitialization strategies. For this initial assessment, we use the permuted MNIST problem (Goodfellow et al., 2014; Zenke et al., 2017), which consists of several tasks, each corresponding to different random permutations of the pixels of the images of the MNIST dataset. We train networks on 1,000 different permutations with a mini-batch size of 30. For each permutation, we only do one pass through the data, resulting in 2,000 updates to the network per task. We use a feed-forward network with ReLU activations, three hidden layers, and 100 units per layer.

We include two baselines, the network trained without any modification, the *base system*, and the network trained using L2-regularization, the *base system using L2-regularization*. Using L2-regularization has been reported to be a strong baseline in the permuted MNIST problem (Dohare et al., 2024). We add selective weight reinitialization to the base system and compare its performance against the two baselines. We tuned the hyper-parameters of the baselines and selective weight reinitialization using a grid search; see Appendix A for more details on hyper-parameter tuning.

We report the average online accuracy per task of each learning system in Figure 1. We refer to the average online accuracy as the performance for the remainder of this section. The base system (in black in Figure 1) had an initial increase in performance followed by a steady decrease. The base system with L2-regularization (in pink) maintained stable performance throughout training. Selective weight reinitialization with magnitude utility (Figure 1a) experienced a less severe performance drop than the base system, but its performance had a lot of variability. Selective weight reinitialization with gradient utility and reinitialization to the mean (in orange in Figure 1b) had higher performance than the base system, but it still suffered from loss of plasticity. Finally, selective weight reinitialization with gradient utility and reinitialization with initial distribution (in blue) showed higher performance than the base system and experienced no drop in performance. We present further analysis involving correlates of loss of plasticity in Appendix B.

**Takeaways.** Regardless of the reinitialization strategy, gradient utility resulted in higher performance than magnitude utility. Between the two variants of selective weight reinitialization that used gradient utility, only the one using reinitialization with initial distribution maintained plasticity. However, it is noteworthy that reinitialization to the mean had better initial performance than reinitialization with initial distribution. Henceforth, we focus only on selective weight reinitialization with gradient utility and report the results with magnitude utility in the appendices.

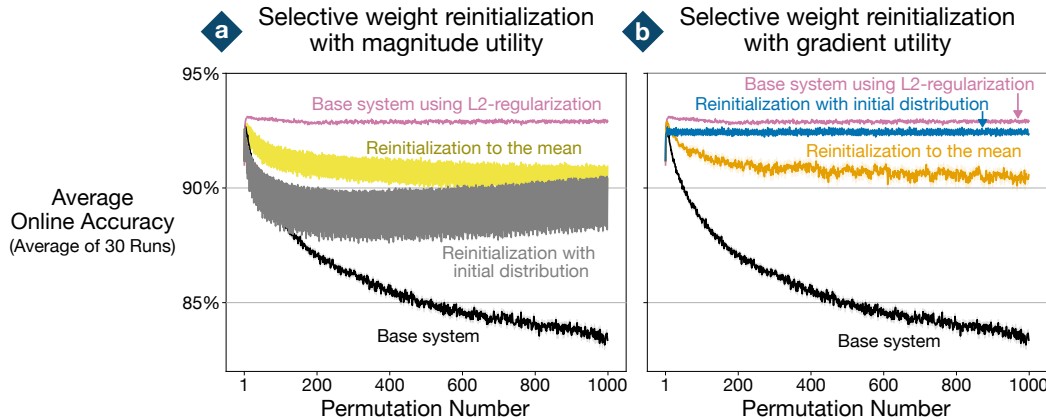

Figure 1: Average online accuracy of selective weight reinitialization with (**a**) magnitude utility and (**b**) gradient utility. Each line is the average of 30 runs while the shaded regions correspond to the standard error. All variants of selective weight reinitialization had higher average online accuracy than the base system. However, only selective weight reinitialization with gradient utility and reinitialization with initial distribution completely maintained plasticity throughout the experiment.

## 4 REINITIALIZING WEIGHTS VS REINITIALIZING UNITS FOR MAINTAINING PLASTICITY IN FEED-FORWARD NETWORKS

We proceed to use selective weight reinitialization and continual backpropagation to study the question: are there settings where reinitializing weights is more effective at maintaining plasticity than reinitializing units? Both algorithms work similarly but at different levels in the network. Continual backpropagation (CBP) reinitializes low-utility units in the network according to a replacement rate. Moreover, newly reinitialized units are protected from being reinitialized again for a number of updates until the units have met a maturity threshold. On the other hand, selective weight reinitialization (SWR) reinitializes a proportion of low-utility weights every certain number of steps.

We devised two settings where reinitializing units may fail to balance maintaining plasticity and preventing the loss of previously learned information. The first setting is when the network architecture includes layer normalization. Layer normalization is a technique for normalizing the values of units in a layer by subtracting the sample average and dividing by the sample standard deviation (Ba et al., 2016). We suspect that reinitializing units may affect the statistics used in layer normalization, harming performance.

The second setting where we expect reinitializing units to be less effective is when the network has a small number of units. In such a case, reinitializing even a single unit may modify a large portion of the network at once. For example, reinitializing a single unit in a network with 100 units changes 1% of the weights, whereas the change would be ten times as large in a network with only ten units. In either case, reinitializing weights allows for a smaller portion of the network to be modified because it works at a lower level of granularity.

We use four network architectures in the permuted MNIST problem to assess the effectiveness of reinitializing units vs weights. First, we use a feed-forward network with three hidden layers, ReLU activations, and 100 units per layer, *large network* setting. Second, we add layer normalization to the large network, *large network with layer norm* setting, to see if reinitializing units or weights affect the statistics in the layer normalization modules and whether that affects performance. Third, we use the same network architecture but with ten units per layer, *small network* setting, to assess the effectiveness of reinitializing units for maintaining plasticity in small networks. Finally, we add layer normalization to the small network, *small network with layer norm* setting, to test how the two settings interact. We use the online accuracy averaged over each task as a performance measure.

For each architecture, we present the performance of five learning systems:

1. the network architecture trained without any modification, *base system* (black lines in Figure 2),

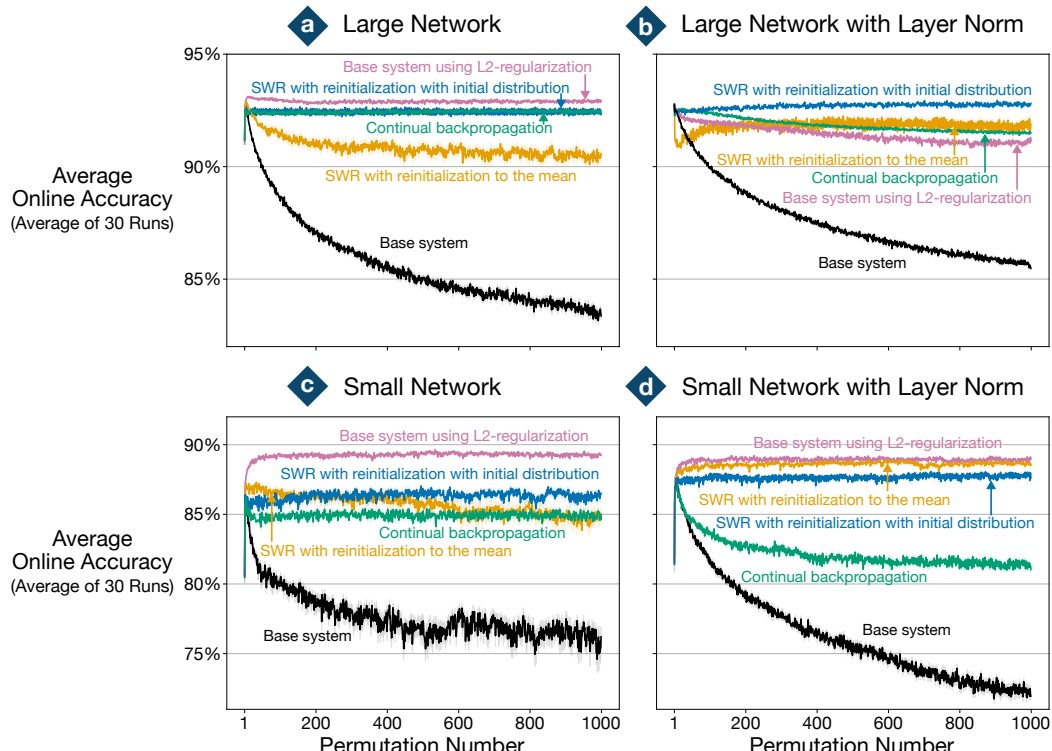

Figure 2: Average online accuracy of selective weight reinitialization and continual backpropagation in four settings: (**a**) large network, (**b**) large network with layer norm, (**c**) small network, and (**d**) small network with layer norm. Each line is the average of 30 runs; the shaded regions correspond to the standard error. Selective weight reinitialization with initialization with initial distribution maintained plasticity in all four settings, whereas continual backpropagation maintained plasticity only in the large network setting.

2. the network trained using L2-regularization, *base system with L2-regularization* (pink),

3. the network trained with CBP, *continual backpropagation* (green),

4. the network trained with SWR with gradient utility and reinitialization with initial distribution, *SWR with reinitialization with initial distribution* (blue),

5. and SWR with gradient utility and reinitialization to the mean, *SWR with reinitialization to the mean* (orange).

We used a grid search to tune the hyper-parameters of each learning system for each architecture. We include more details about hyper-parameter selection along with the results of SWR with magnitude utility in Appendix A.

**Effectiveness of reinitialization schemes when using layer normalization.** Contrasting the results without and with layer norm (left and right columns of Figure 2, respectively), we confirm our initial intuition: reinitializing units is less effective at maintaining plasticity when the network employs layer normalization. When using a large network, using CBP resulted in stable performance without layer normalization (Figure 2a), but resulted in steadily decreasing performance when using layer normalization (Figure 2b). In the small network, using CBP resulted in a drop in performance in the networks with and without layer normalization (bottom row of Figure 2); this effect was more severe when using layer normalization (Figure 2d). On the other hand, SWR with initialization with initial distribution resulted in stable performance in the networks with and without layer normalization.

The performance drop in CBP is partially explained by the change in sample average and standard deviation after reinitializing a unit. Table 1 shows the absolute change in the statistics of the activations before and after a reinitialization step. In the small network setting, reinitializing units caused a larger change in sample average and standard deviation than reinitializing weights. In the large network setting, there was no consistent pattern across all the layers in the network. Notably, the loss

Table 1: Average per task of the absolute difference in sample average (Avg) and standard deviation (SD) of activations per layer after a reinitialization step in CBP or SWR with reinitialization with initial distribution. The quantities reported are averages of 30 runs. The standard error of each measurement was less than 0.002 in the large network setting and less than 0.03 in the small network setting.

| | **Large Network Setting** | | | | | | **Small Network Setting** | | | | | |
| | Change in Avg | | | Change in SD | | | Change in Avg | | | Change in SD | | |
| **Layer** | 1 | 2 | 3 | 1 | 2 | 3 | 1 | 2 | 3 | 1 | 2 | 3 |
| CBP | 0.35 | 0.04 | 0.02 | 0.21 | 0.04 | 0.02 | 1.09 | 3.48 | 2.96 | 0.88 | 1.63 | 2.69 |
| SWR | 0.06 | 0.07 | 0.05 | 0.11 | 0.08 | 0.21 | 0.38 | 0.72 | 0.65 | 0.36 | 0.4 | 0.44 |

of plasticity when reinitializing units was more severe in the small network with layer norm than in the large network with layer norm. We explore other explanations for the difference in performance in Appendix B.

**Effectiveness of reinitialization schemes in small networks**. We confirm our initial intuition that reinitializing units is less effective than reinitializing weights in networks with few units. We contrast the results in large and small networks (top and bottom rows of Figure 2, respectively). In the small network setting (Figure 2c), the performance of CBP decreased after the first few tasks, but performance stabilized soon after. In contrast, the performance of CBP was stable in the large network without layer normalization (Figure 2a). When using layer normalization, CBP experienced a steady decrease in performance in both the large and small networks (Figures 2b and 2d, respectively), but the effect was more severe in the small network.

One possible explanation for the difference in CBP's and SWR's performance is that SWR can reinitialize weights at a slower rate because it works at the weights' level. This was true in the small network setting, where CBP reinitialized weights at a rate of 8.35 per parameter update, whereas SWR with reinitialization with initial distribution reinitialized 1.36 weights per update (see Appendix A for a detailed calculation of this values). However, in the small network with layer norm setting, CBP reinitialized 0.084 weights per update, whereas SWR reinitialized 0.687 weights per update. Thus, the reinitialization rate is not entirely responsible for the difference in performance. We explore other possible explanations for the difference in performance by looking into the correlates of loss of plasticity in Appendix B.

**Takeaways.** We found two settings in which reinitializing units was less effective at maintaining plasticity than reinitializing weights: when the network uses layer normalization and when it has a small number of units. The first setting is particularly relevant for modern applications because layer normalization has become the standard approach for normalizing activations in transformer architectures (Vaswani et al., 2017; Devlin et al., 2018; Dosovitskiy et al., 2021), the architectures responsible for the success of large language models. The second setting is relevant for continual learning. If one subscribes to the big world hypothesis (Javed & Sutton, 2024), which poses that a learning system should be orders of magnitude smaller than the world they are learning about, then one can no longer rely on the size of the network to design successful learning algorithms. Working under the big world hypothesis, reinitializing weights is more effective than reinitializing units because it does not rely on having a large number of units to maintain plasticity. Finally, although L2-regularization was a strong baseline in this problem, it is not always sufficient for maintaining plasticity. This can already be seen in Figure 2b, but we also make the same observation in the next section in a more complex problem.

## 5 REINITIALIZING WEIGHTS VS REINITIALIZING UNITS FOR MAINTAINING PLASTICITY IN RESNET-18 AND VISION TRANSFORMERS

We move on to a large-scale demonstration with real-world data. This section aims to compare the effectiveness of reinitializing weights for maintaining plasticity in a more realistic dataset using

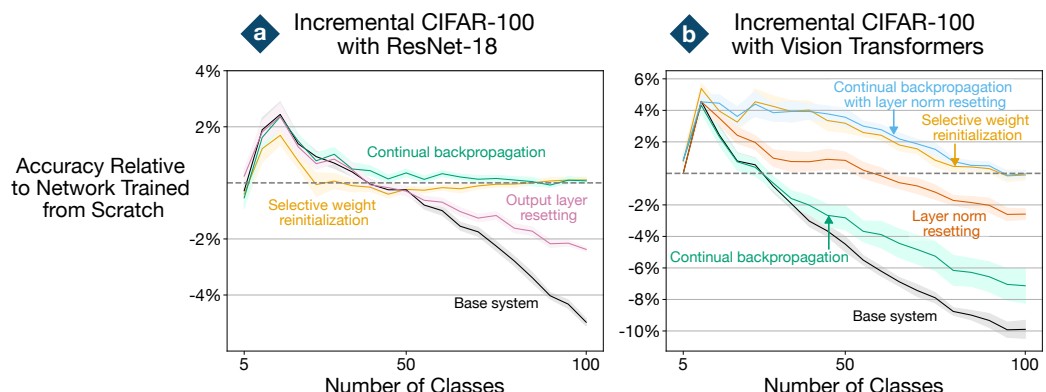

Figure 3: Accuracy relative to a network trained from scratch for each learning system in CIFAR-100 with (**a**) ResNet-18 and (**b**) vision transformers. Each line is the average of 15 runs in the ResNet-18 plot and 10 runs in the vision transformer plot; the shaded regions correspond to the standard error. Continual backpropagation and selective weight reinitialization both maintain plasticity in ResNet-18. In vision transformers, only selective weight reinitialization maintains plasticity. However, continual backpropagation matches the performance of selective weight reinitialization when combined with layer norm resetting.

modern architectures. Our secondary goal is to determine if reinitializing weights still shows an advantage over reinitializing units for maintaining plasticity in more complex problems.

We use the class incremental CIFAR-100 problem studied in by Chaudhry et al. (2018) and Dohare et al. (2024). In this problem, networks are trained on an increasing number of classes from the CIFAR-100 dataset, which consists of 100 classes with 500 training and 100 test images per class. In the first task, the network is trained to predict five classes. After several epochs, the number of classes in the dataset increases by five, and a new task begins. This process continues until the dataset contains all 100 classes, resulting in 20 tasks.

To isolate the loss of plasticity effect, we implement measures for preventing forgetting and overfitting. To prevent forgetting, the network is constantly retrained on old classes; new classes are added to the dataset, but old classes are not removed. To prevent overfitting, we use image transformations and early stopping. For image transformation, we use random cropping with padding of 4 pixels on each side of the image, random horizontal flipping with 0.5 probability, and random rotation between 0 and 15 degrees. To implement early stopping, we train the network for a fixed number of epochs, measure its accuracy after each epoch on a validation dataset of 50 images per class taken from the training set, and reset the network to the network with the highest validation accuracy at the start of each task. The measure of performance is the highest test accuracy achieved during each task.

We used two network architectures for this problem: ResNet-18 and vision transformers. Both architectures are trained using stochastic gradient descent with a momentum of 0.9. The ResNet-18 architecture was trained for 200 epochs per task and used a learning rate scheduler that decreased the learning rate at epochs 60, 120, and 160. The vision transformer architecture was trained for 100 epochs per task and used a linear learning rate schedule. The learning rate increased to 0.01 for the first 30 epochs and then decreased to zero during the last 70 epochs. We used fewer training epochs for the vision transformer since we did not notice any increase in performance when using a larger number of epochs. For both architectures, the learning rate scheduler was restarted at the start of each task. The ResNet-18 architecture used batch normalization (Ioffe & Szegedy, 2015). The vision transformer architecture used dropout and layer normalization. Both architectures used L2-regularization.

We compare the performance of continual backpropagation and selective weight reinitialization when combined with these architectures. For the ResNet-18, we compare to the results presented by Dohare et al. (2024). For the vision transformer, we apply continual backpropagation only between feed-forward layers in the network since an extension of continual backpropagation for attention layers has yet to be proposed. We apply selective weight reinitialization to all the weight matrices and bias vectors in the architectures. During hyper-parameter tuning, we found that selective

weight reinitialization with reinitialization to the mean was more effective in these architectures. See Appendix C for more details on hyper-parameter selection.

We present the difference in the performance of each learning system compared to a network trained from scratch on the same set of classes. Additionally, we include other reinitialization baselines. We include a baseline that reinitializes the output layer in the ResNet-18 architecture and another baseline that reinitializes the layer norm parameters in the vision transformer architecture. We did not notice any increase in performance from reinitializing the parameters in the batch normalization layers in ResNet-18 or reinitializing the output layer in vision transformers, so we omitted those baselines. For these reinitialization baselines, reinitialization happened before the start of each new task.

In ResNet-18, continual backpropagation and selective weight reinitialization maintained plasticity (Figure 3a). However, continual backpropagation scores a higher test accuracy than selective weight reinitialization over most tasks during the experiment. In vision transformers, selective weight reinitialization maintains plasticity, whereas continual backpropagation slightly improves over the base system (Figure 3b). However, when combined with layer norm resetting, continual backpropagation performs just as well as selective weight reinitialization; see Appendix C for more details about the success of layer norm resetting. It is important to note that continual backpropagation with layer norm resetting has privileged knowledge of when tasks change, whereas selective weight reinitialization does not use that information.

**Takeaways.** The results in ResNet-18 show that reinitializing weights is a viable strategy for maintaining plasticity, albeit with lower generalization performance than reinitializing units. On the other hand, the results in vision transformers confirm that reinitializing units is less effective than reinitializing weights when the architecture includes layer normalization. Nevertheless, combining layer norm resetting and reinitializing units is an effective technique for maintaining plasticity in vision transformers. An interesting observation in vision transformers with continual backpropagation is that attention layers do not seem to be the source of loss of plasticity. Continual backpropagation was only used between feed-forward layers, ignoring attention layers. Yet, continual backpropagation maintains plasticity when combined with layer norm resetting.

## 6 CONCLUSION AND FUTURE WORK

We presented an algorithm for reinitializing weights for maintaining plasticity, a reinitialization scheme that had remained unexplored in the loss of plasticity literature. Through comparisons in continual supervised learning, we uncovered two settings where reinitializing weights is more effective at maintaining plasticity than reinitializing units. Moreover, the idea of reinitializing weights is easier to implement than reinitializing units since it does not have to account for the complex interconnections between the structures in the network, which is a difficulty also encountered in structural pruning (Fang et al., 2023). Finally, we demonstrated in a class-incremental problem that reinitializing weights maintains plasticity in larger architectures that employ many of the techniques used in modern applications.

While effective at maintaining plasticity in various settings, selective weight reinitialization has one drawback: no single reinitialization strategy works best in all cases. Specifically, there is no dominant reinitialization strategy. Practitioners would have to test both reinitialization strategies presented in this paper and different values for the reinitialization frequency and proportion to find a configuration that works well in their setting. Finding a reinitialization strategy that works well in every setting would significantly improve the applicability of selective weight reinitialization.

Lastly, as mentioned in the related work section, selective weight reinitialization shares characteristics with dynamic sparse training algorithms but entirely focuses on maintaining plasticity. Adapting selective weight reinitialization to train a sparse network and maintain plasticity is possible. The algorithm would be initialized with a sparse network and, instead of reinitializing weights, it would prune active and regrow inactive connections in the same manner described in Section 3. If implemented in a truly sparse fashion, this algorithm could result in fast learning while maintaining plasticity.

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

# A  ADDITIONAL RESULTS AND DETAILS ABOUT PERMUTED MNIST EXPERIMENTS

**Hyper-parameter tuning.** For the experiments in Permuted MNIST presented in Sections 3 and 4, we tuned the hyper-parameters of each learning system using a grid search with ten runs per hyper-parameter combination. After the search, we selected the hyper-parameter values that resulted in the highest average online accuracy throughout the training period, i.e., the area under the curve. For the base systems, we tuned the learning rate parameter, $\alpha$. For the base systems using L2-regularization, we tuned the regularization factor. For continual backpropagation, we tuned the replacement rate, $rr$, and maturity threshold, $mt$. For selective weight reinitialization, we tuned the reinitialization frequency, $\tau$, and proportion, $p$. The base systems using L2-regularization, the continual backprop-agation systems, and the selective weight reinitialization systems used the same learning rate as the base systems in the corresponding setting. Finally, for the layer normalization settings, we compared layer norm before and after the activation; using layer norm after the activation resulted in higher average online accuracy in both the small and large networks.

Table 2 shows the hyper-parameter values tested for each algorithm. Underlined values correspond to the values used in the main text, except for selective weight reinitialization. For selective weight reinitialization, we labelled values with GD for gradient utility with reinitialization with initial distribution, GM for gradient utility with reinitialization to the mean, MD for magnitude utility with reinitialization with initial distribution, and MM for magnitude utility with reinitialization to the mean to indicate the values used for each of the corresponding algorithms in the main paper.

**Selective weight reinitialization with magnitude utility in permuted MNIST.** In Section 4, we omitted the results using selective weight reinitialization with magnitude utility. We present those results in Figure 4. The hyper-parameter values were chosen as described above. In every setting we tested, magnitude utility had lower performance than gradient utility.

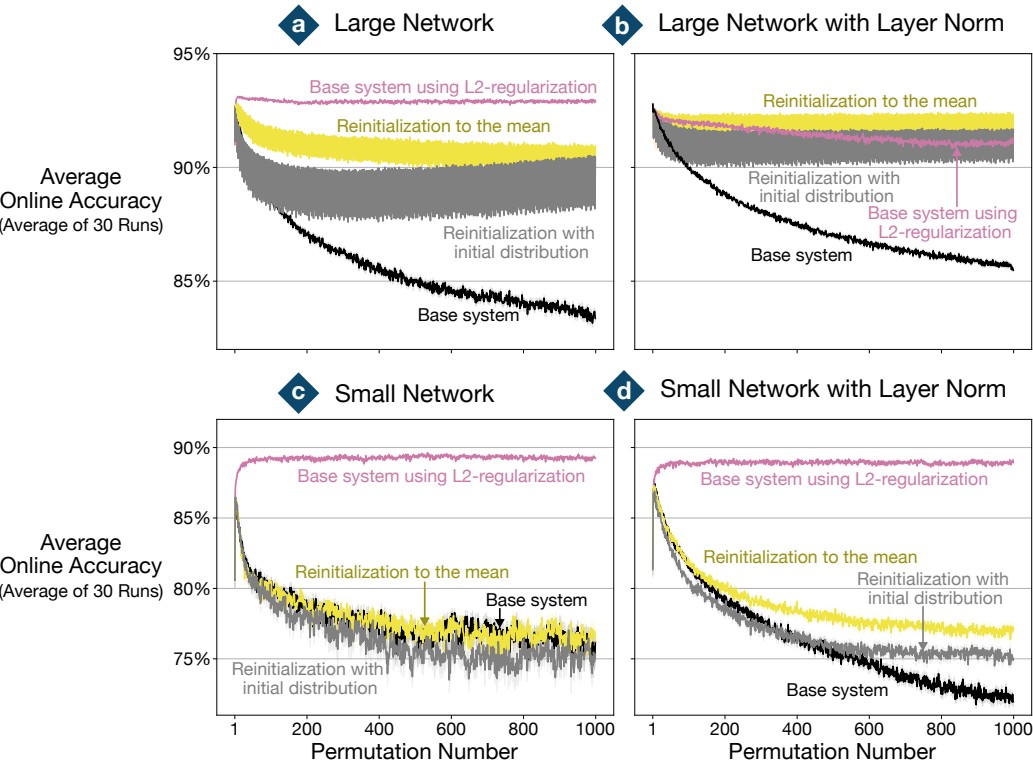

Figure 4: Average online accuracy of selective weight reinitialization with magnitude utility in the (**a**) large network setting, (**b**) large network with layer norm setting, (**c**) small network, and (**d**) small network with layer norm. Gradient utility achieved higher performance than magnitude utility in all four settings.

Table 2: Hyper-parameter values used in each of the four network settings in Permuted MNIST.

| **Large Network Setting** | |
| --- | --- |
| Base system | $\alpha \in \{0.1, \underline{0.05}, 0.01, 0.005, 0.001, 0.0005\}$ |
| Base system using L2-regularization | L2-factor of $1 \times 10^{\beta}$ with $\beta \in \{-3, \underline{-4}, -5, -6, -7, -8\}$ |
| Continual backpropagation | $rr \in \{1 \times 10^{-1}, 1 \times 10^{-2}, 1 \times 10^{-3}, \underline{1 \times 10^{-4}}, 1 \times 10^{-5}\}$ 
 $mt \in \{\underline{1}, 5, 50, 100, 500\}$ |
| Selective weight reinitialization | $\tau \in \{300, 600, 1200^{\text{GM}}, 2400^{\text{GD}}, 4800^{\text{MD, MM}}\}$ 
 $p \in \{0.05, 0.1, 0.2^{\text{GM}}, 0.4^{\text{GD}}, 0.8^{\text{MD, MM}}\}$ |

| **Large Network with Layer Norm Setting** | |
| --- | --- |
| Base system | $\alpha \in \{0.5, \underline{0.1}, 0.05, 0.01, 0.005, 0.001, 0.0005\}$ |
| Base system using L2-regularization | L2-factor of $1 \times 10^{\beta}$ with $\beta \in \{-3, -4, \underline{-5}, -6, -7, -8\}$ |
| Continual backpropagation | $rr \in \{1 \times 10^{-1}, \underline{1 \times 10^{-2}}, 1 \times 10^{-3}, 1 \times 10^{-4}, 1 \times 10^{-5}\}$ 
 $mt \in \{\underline{1}, 5, 50, 100, 500\}$ |
| Selective weight reinitialization | $\tau \in \{300, 600, 1200^{\text{GD}}, 2400^{\text{GM}}, 4800^{\text{MD, MM}}\}$ 
 $p \in \{0.05, 0.1^{\text{GD}}, 0.2, 0.4, 0.8^{\text{MD, MM, GM}}\}$ |

| **Small Network Setting** | |
| --- | --- |
| Base system | $\alpha \in \{0.1, \underline{0.05}, 0.01, 0.005, 0.001, 0.0005\}$ |
| Base system using L2-regularization | L2-factor of $1 \times 10^{\beta}$ with $\beta \in \{-3, \underline{-4}, -5, -6, -7, -8\}$ |
| Continual backpropagation | $rr \in \{1 \times 10^{-1}, 1 \times 10^{-2}, \underline{1 \times 10^{-3}}, 1 \times 10^{-4}, 1 \times 10^{-5}\}$ 
 $mt \in \{1, \underline{5}, 50, 100, 500\}$ |
| Selective weight reinitialization | $\tau \in \{75, 150, 300^{\text{MM}}, 600^{\text{GM}}, 1200^{\text{MD, GD}}\}$ 
 $p \in \{0.005^{\text{MM,MD}}, 0.01, 0.05, 0.1^{\text{GM}}, 0.2^{\text{GD}}\}$ |

| **Small Network with Layer Norm Setting** | |
| --- | --- |
| Base system | $\alpha \in \{0.5, \underline{0.1}, 0.05, 0.01, 0.005, 0.001, 0.0005\}$ |
| Base system using L2-regularization | L2-factor of $1 \times 10^{\beta}$ with $\beta \in \{-3, \underline{-4}, -5, -6, -7, -8\}$ |
| Continual backpropagation | $rr \in \{1 \times 10^{-1}, 1 \times 10^{-2}, 1 \times 10^{-3}, 1 \times 10^{-4}, \underline{1 \times 10^{-5}}\}$ 
 $mt \in \{\underline{1}, 5, 50, 100, 500\}$ |
| Selective weight reinitialization | $\tau \in \{75, 150, 300^{\text{GM}}, 600, 1200^{\text{MM, MD, GD}}\}$ 
 $p \in \{0.005, 0.01, 0.05, 0.1^{\text{GM, GD}}, 0.2^{\text{MM, MD}}\}$ |

**Notes on implementation of continual backpropagation.** We used the implementation provided by Dohare et al. (2024), the original authors of the continual backpropagation algorithm. In this implementation, the layer normalization parameters associated with a hidden unit are reinitialized along with the corresponding unit. Additionally, the implementation uses contribution utility computed from the data in the current mini-batch instead of as a running average. We chose not to use

running averages to reduce the number of tunable hyper-parameters and because the mini-batch size was large enough to provide accurate estimates for computing the contribution utility.

**Computing the reinitialization rate of continual backpropagation and selective weight reinitialization.** Here, we explain how to compute the quantities reported in Section 4. In the case of selective weight reinitialization, the number of weights reinitialized per parameter update is simply the number of parameters in the network times the reinitialization proportion divided by the reinitialization frequency. For the small network setting, the network contained 8,180 parameters, and selective weight reinitialization with reinitialization with initial distribution used a reinitialization proportion of 0.2 and a reinitialization frequency of 1,200. Thus, the reinitialization rate was 1.36. For the small network with layer norm setting, the network contained 8,240 parameters, the reinitialization proportion was 0.1, and the reinitialization frequency was 1,200, resulting in a reinitialization rate of 0.687.

In the case of continual backpropagation, computing the reinitialization rate for weights is more complicated. In the small network setting, continual backpropagation used a replacement rate of $1 \times 10^{-3}$. Since the network contains ten units per layer, continual backpropagation reinitializes one unit every 100 steps. In the first layer, reinitializing a unit is equivalent to reinitializing one row of the input weight matrix (784 weights), one bias term (1 weight), and a column in the output weight matrix of the unit (10 weights), resulting in 795 weights reinitialized. In the second layer, we use the same formula minus one because one of the entries in the row was reinitialized when reinitializing the previous layer, which results in 20 weights reinitialized (10 from the input weight matrix, ten from the output weight matrix, one from the bias term, and -1 from reinitializing weights in the previous layer), and the same in the third layer. Thus, continual backpropagation reinitializes 835 weights every 100 updates, equivalent to 8.35 weights per update. When using layer norm, we add plus two to the weights in each layer since the layer norm parameters are also reinitialized. In that setting, continual backprop used a replacement rate of $1 \times 10^{-5}$, or one unit every 10,000 updates. Thus, continual backpropagation reinitializes 841 weights every 10,000 updates or 0.0841 weights per update.

## B    CORRELATES OF LOSS OF PLASTICITY IN PERMUTED MNIST

Here, we present additional measurements that could explain the performance of the learning systems presented in the main text. Our goal is to rule out pathological scenarios that often occur along with loss of plasticity but are not the root causes of it. We look for three pathological scenarios. First, we look for a large accumulation of dead units corresponding to a loss of representational capacity in the network. We measure the percent of units that always output zero on a random sample of 2,000 MNIST images after a new permutation is applied but before training recommences. Second, we look for significant increases in the average magnitude of the parameters of the network, which could signal instability in the optimization process. We measure the average magnitude of the weight at the end of each task. Lastly, we look for shrinkage of the average magnitude of the gradients, which could signal a drastic slowdown in learning. We measure the average gradient magnitude online as the network learns from new observations.

**Correlates of loss of plasticity in the initial assessment.** In Section 3, we presented comparisons between selective weight reinitialization using two different utility functions. While we noticed differences in performance, we did not delve deeper into the qualitative difference between the two. Figure 5 shows the three correlates of loss of plasticity. The base system showed every pathological scenario we described; it had many dead units, increasing weight magnitude, and decreasing gradient magnitude. On the other hand, selective weight reinitialization with gradient utility and reinitialization with initial distribution (blue lines in Figure 5) avoided all of these scenarios. Yet, the results also show that these measurements are unreliable at predicting loss of plasticity. Selective weight reinitialization with gradient utility and reinitialization to the mean also showed the same pathological scenarios as the base system. Nevertheless, it had a higher and more stable performance than selective weight reinitialization using magnitude utility, which scored better in these three measurements.

**Correlates of loss of plasticity in the large network with layer norm setting.** In Section 4, we proposed that reinitializing units drastically changed the statistics used in layer normalization modules. We verified that was the case for the small network with layer normalization setting but

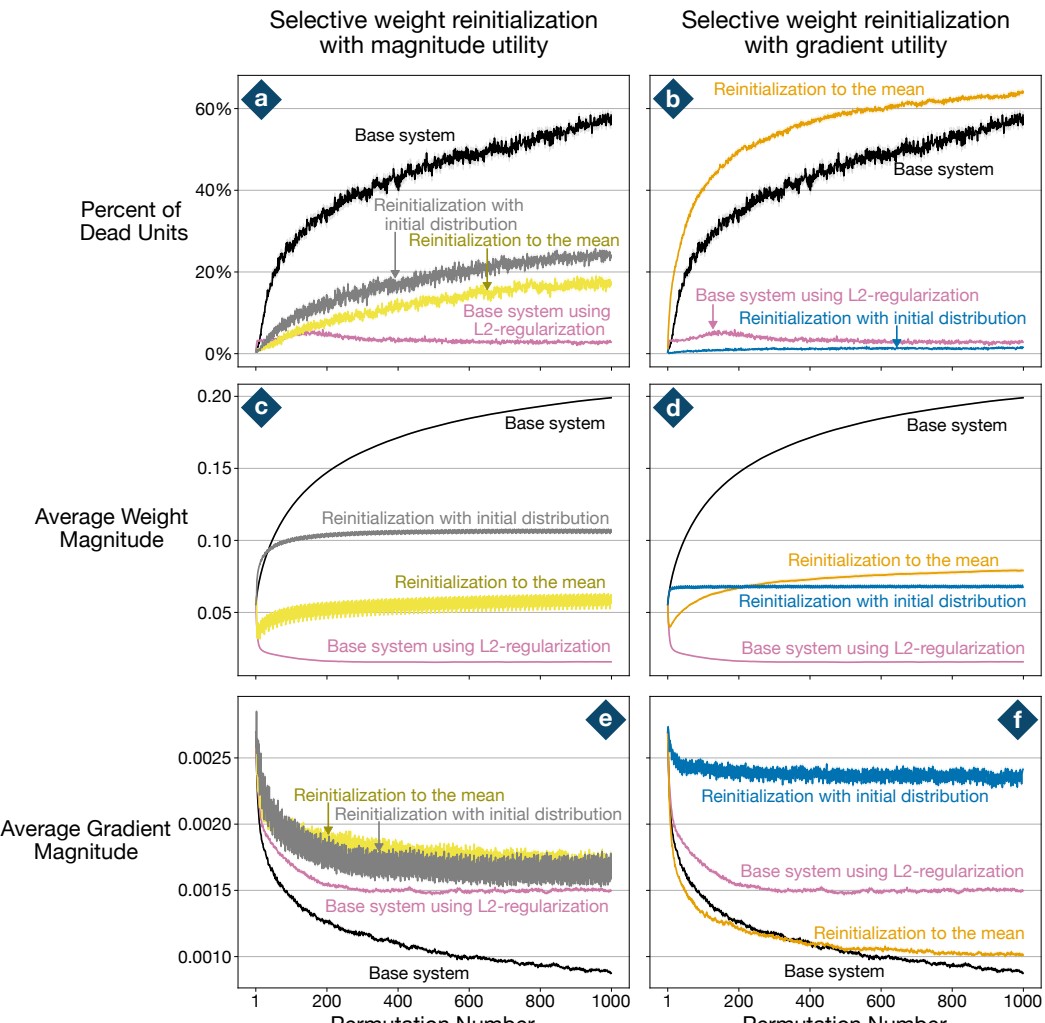

Figure 5: Correlates of loss of plasticity for the initial assessment presented in Figure 1. Each line corresponds to the average of 30 runs, whereas the shaded region corresponds to one standard error. The base systems showed a large percentage of dead units, increasing weight magnitude, and decreasing gradient magnitude, which could explain its poor performance. Selective weight reinitialization with gradient utility and reinitialization with initial distribution avoided these three scenarios and maintained plasticity throughout the experiment.

did not find the same effect when using a large network. Here, we look deeper into the network to verify if reinitializing units resulted in other pathological scenarios.

Figure 6 shows the correlates of loss of plasticity. Only the correlates for selective weight reinitialization with gradient utility are shown. Continual backpropagation maintained a small percent of dead units and a large average gradient but also saw an increase in weight magnitude. This could explain why it prevented some loss of plasticity; it prevented two of the three pathological scenarios. The measurements corresponding to selective weight reinitialization with reinitialization to the mean are puzzling. It accumulated a large percentage of dead units and saw a large decrease in gradient magnitude. Yet, its performance was higher than continual backpropagation and the base system using L2-regularization by the end of the experiment.

**Correlates of loss of plasticity in the small network with layer norm setting.** In Section 4, we proposed that reinitializing units would result in more weights being reinitialized on average. This was only true when not using layer normalization, which suggests other explanations for the poor performance of continual backpropagation in small networks with layer norm setting. We look at the correlates of loss of plasticity for such an explanation.

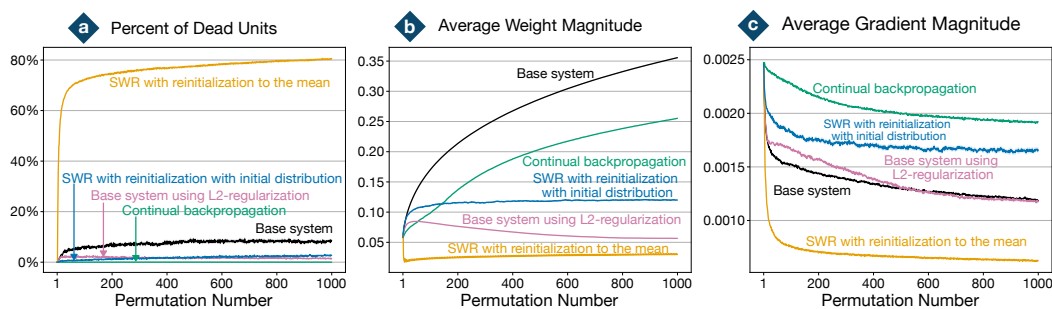

Figure 6: Correlates of loss of plasticity for a large network with layer norm setting presented in Figure 2b. Each line corresponds to the average of 30 runs, whereas the shaded region corresponds to one standard error. Continual backpropagation prevents a large accumulation of dead units and a decrease in gradient magnitude, but it sees an increase in the average weight magnitude, which could explain its poor performance.

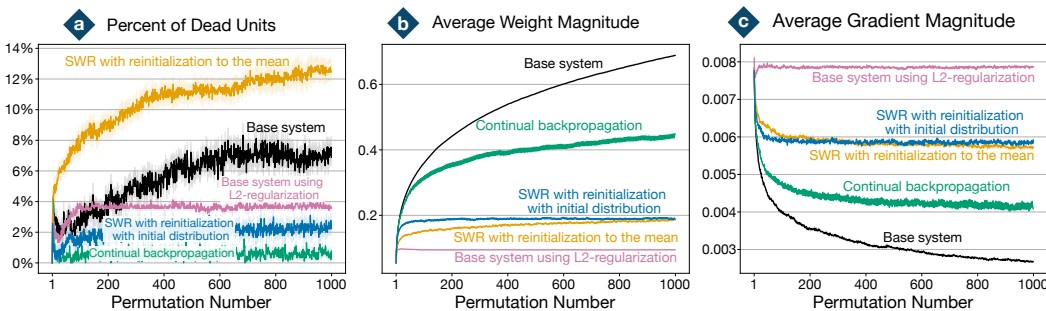

Figure 7: Correlates of loss of plasticity for small network with layer norm setting presented in Figure 2d. Each line corresponds to the average of 30 runs, whereas the shaded region corresponds to one standard error.

Once again, we found that continual backpropagation was very effective at keeping units alive (Figure 7a). However, continual backpropagation also had an increasing average weight magnitude and a decreasing average gradient magnitude. Along with the results in Figure 6, these findings suggest that addressing the dormant neuron problem is not enough to prevent the loss of plasticity, which contrasts the results found by Sokar et al. (2023).

## C ADDITIONAL RESULTS AND DETAILS ABOUT CLASS-INCREMENTAL CIFAR-100 EXPERIMENTS

**Architectures.** For the ResNet-18 experiments, we used the same architecture, hyper-parameters, and implementation used by Dohare et al. (2024). For the vision transformer experiments, we modified the implementation provided in the torchvision python package (maintainers & contributors, 2016) to include the continual backpropagation implementation from Dohare et al. (2024) between feed-forward layers. We tried several architecture settings to maximize the test accuracy on the CIFAR-100 dataset during 100 epochs of training using stochastic gradient descent with a learning rate of 0.01, a dropout probability of 0.1, and a momentum of 0.9. The best configuration we found was a patch size of 4, 8 encoder blocks with an embedding dimension of 384 each, 12 attention heads per block, and 1,536 hidden units in the multi-layer perceptron block. With this configuration, the architecture has 14,279,140 parameters.

**Hyper-parameter tuning in vision transformers.** Except for the selective weight reinitialization results, we used the same results presented by Dohare et al. (2024). For selective weight reinitialization, we first tested hyper-parameter values randomly to find a suitable range for the grid search. Then, we tried reinitialization frequencies in $\{130, 260, 520, 1040\}$ and reinitialization proportions in $\{0.025, 0.05, 0.1\}$. We ran each parameter combination for five different random seeds. We selected the combination that maximized the sum of the highest test accuracy per task in the class-incremental CIFAR-100 problem, equivalent to the area under the curve of the lines in Figure 8a.

The main text shows the results of selective weight reinitialization with reinitialization frequency and proportion of 260 and 0.05, respectively.

**Hyper-parameter tuning in vision transformers.** For the base system, we tested values for the learning rate in $\{0.2, 0.1, 0.05, 0.01, 0.005\}$, values for the L2-regularization factor in $\{1 \times 10^{-4}, 1 \times 10^{-5}, 5 \times 10^{-6}, 2 \times 10^{-6}, 1 \times 10^{-6}, 1 \times 10^{-7}, 1 \times 10^{-8}\}$, and values for the dropout probability in $\{0.0, 0.05, 0.1, 0.15\}$. We did not scale the L2-regularization factor by the learning rate to keep the regularization strength constant even as the learning rate decreased to zero due to the scheduler. In the main text, we used a learning rate of $0.01$, an L2-regularization factor of $2 \times 10^{-6}$, and a dropout probability of $0.1$. These values were selected to maximize the test accuracy in the CIFAR-100 problem during 100 epochs of training. The network was trained using stochastic gradient descent with a momentum of 0.9 and a mini-batch size of 90. All the other systems use the same learning rate, L2-regularization factor, dropout probability, and momentum term as the base system.

For continual backpropagation with vision transformers, we tested values for the replacement rate in $\{1 \times 10^{-4}, 1 \times 10^{-5}, 1 \times 10^{-6}, 1 \times 10^{-7}\}$ and maturity threshold in $\{100, 1000, 10000\}$ using five random seeds per parameter combination. We selected the combination that maximized the sum of the highest test accuracy per task, equivalent to the area under the curve of the lines in Figure 8b. In the main text, we present the results of continual backpropagation using a replacement rate of $1 \times 10^{-6}$ and a maturity threshold of 100. We used the contribution utility computed on the current mini-batch of data.

After an initial random search, we tested reinitialization frequencies in $\{30, 65, 130\}$ and reinitialization proportions in $\{0.005, 0.01, 0.02\}$ for selective weight reinitialization. The main text shows the results of selective weight reinitialization with reinitialization frequency and proportion of 65 and 0.01, respectively. We used gradient utility and reinitialization to the mean. Note that reinitialization to the mean reinitializes weights and bias to zero for every parameter matrix and vector, except for the weights in the layer normalization modules. The weights of layer normalization modules were reinitialized to one.

We used the same hyper-parameters as the base system for the layer norm resetting baseline. For continual backpropagation with layer norm resetting, we used the same hyper-parameters as continual backpropagation.

**Test accuracy plot.** The main text presented the accuracy relative to the network trained from scratch. For completeness, we presented the highest test accuracy per task of each algorithm in Figure 8. Figure 3 was created by taking the difference between the performance of each learning system and the network trained from scratch baseline (gray) in Figure 8.

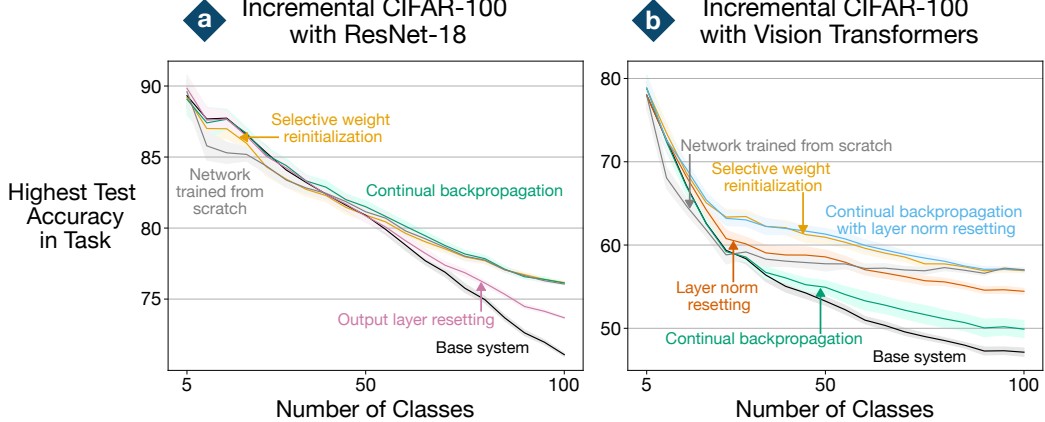

Figure 8: Best test accuracy per task in class-incremental CIFAR-100 with (**a**) ResNet-18 and (**b**) vision transformers. Each line is the average of 15 runs in the ResNet-18 plot and 10 runs in the vision transformer plot; the shaded regions correspond to the standard error.

**Observation about layer norm parameters.** During our experiments, we noticed that the scaling parameter in the layer normalization modules was shrinking in each consecutive task. As a reminder,

the layer normalization performs the following operation,

$$y = \frac{x - \mathbb{E}[x]}{\sqrt{\mathbb{V}[x] + \epsilon}} \cdot \gamma + \beta,$$

where $x$ is an activation in a layer, $\mathbb{E}[x]$ is the sample average of all the activations in the layer, $\mathbb{V}[x]$ is the sample variance, $\epsilon$ is a small positive number to prevent division by zero, and $\gamma$ and $\beta$ are learnable parameters. We found $\gamma$ was shrinking throughout training, a potential form of failure for layer norm. If $\gamma$ reaches zero, then the network stops propagating gradients back to the layers preceding the layer normalization module. This is why we devised the layer norm resetting baseline in Figure 3. Learning systems that maintained plasticity also kept the value of $\gamma$ relatively high (Figure 9). The shrinkage of the scaling factor in layer normalization is an understudied failure mode that deserves a more thorough investigation because of the essential role of layer normalization in large language models.

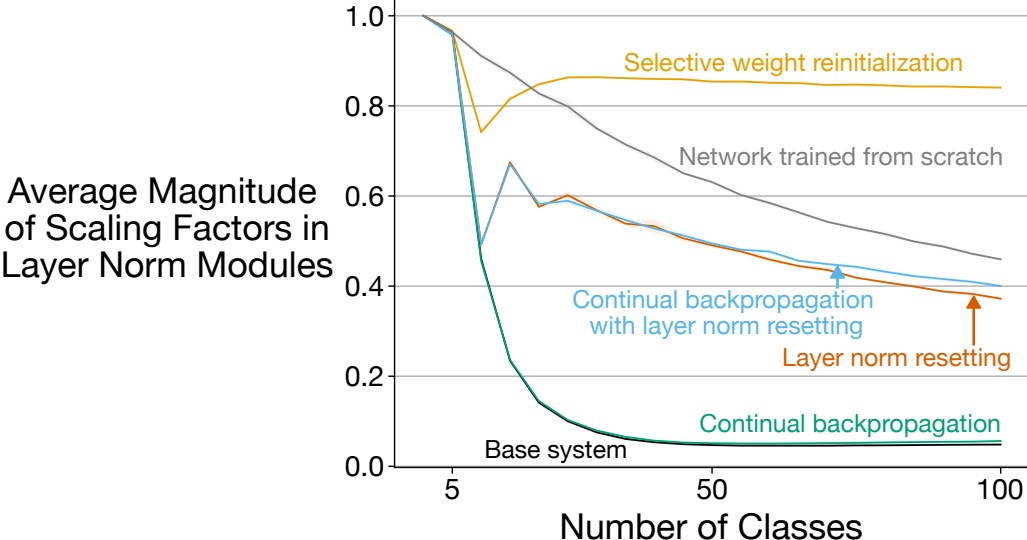

Figure 9: Average magnitude of the learnable parameter $\gamma$ over all the layer normalization modules in the vision transformer architecture. A drastic decrease in the magnitude of $\gamma$ corresponds to a reduction in accuracy in Figure 5.

