# OpenReview forum: "Reinitializing weights vs hidden units for maintaining plasticity in neural networks"
_ICLR.cc/2025/Conference — ICLR 2025 Conference Withdrawn Submission_

### Official Review · Reviewer_S11o · 2024-10-21

**Soundness:** 3
**Presentation:** 3
**Contribution:** 1
**Rating:** 5
**Confidence:** 3

**Summary:**

This work presents a novel method for re-initializing weights of a neural network during continual learning tasks in order to maintain network plasticity. In particular, a multi-stage process is described in which the utility (importance) of network weights is measured, and the lowest utility weights replaced with re-initialized weights at specific ratios and in particular intervals. Utility is measured in two ways, both based upon existing proposals for measurement of parameter importance in network pruning literature. The re-initialization of weights is also achieved in two different manners, one by a random value (based upon the initial parameter initialization process) or based upon a constant value (the mean of the initial layer weights). Performance of this method, as well as a few baselines, is shown to be highly dependent upon network size as well as whether layer normalization is present in the network architecture. It is concluded that weight reinitialization can work well (in particular cases) for plasticity maintenance in neural networks applied to continual learning tasks.

**Strengths:**

- The paper provides a clear overview of the existing literature on network re-initialization as well as the problem at hand. It is also well written in general.
- The method description is well structured and the method appears to combine existing approaches in a novel and simple to implement manner.
- Extensive appendices provide details on the network constructions and architectures, as well as other factors in the success of compared methods.
- The test cases shown appear comprehensive and the method is applied to a range of network scale

**Weaknesses:**

- The depth of the theoretical contribution of this work is limited, especially given that the utility measuring methods are a part of existing literature and the reinitialization methods at a different scale have already been proposed. Therefore the degree of novel contribution is limited.
- Most importantly, it is unclear how to interpret the results given that the proposed method is only capable of outperforming alternative methods in a single demonstrated case. Otherwise existing methods appear to be generally more performant (especially the L2 and CBP methods).
- The take-away sections at each section of results are a good way to focus the reader’s attention, however they do not provide insights of the level that one might desire. For example, it is clear that layer normalization has a major impact upon the performance of continual learning methods. However, the reason as to why this is the case is never fully explored. More focus upon how the results can impact a reader's understanding of network re-initialization would be appreciated..
- The paper compares reinitialization of weights vs reinitialization of nodes but could be clearer about the fact that CBP is in fact a method of reinitialization of nodes - this point is lost within the text.

**Questions:**

The weaknesses section, above, contains a number of concerns to be resolved. Here a few additional questions are posed.
- How would you place your method amongst those in the existing literature? It is clear that there is no claim to state of the art, but what is the message for the broader community of researcher about the relative placement of such a work?
- Are there any methods outside of re-initialization (or the L2 method) which might be appropriate as a baseline? I’m thinking more of methods more of the style of elastic weight consolidation (EWC).
- Why does reinitialization work differently for networks with layer normalization? What property of layer normalization brings about the effects you observe?

---

### Official Review · Reviewer_h8WU · 2024-11-04

**Soundness:** 2
**Presentation:** 2
**Contribution:** 1
**Rating:** 3
**Confidence:** 4

**Summary:**

This work proposed a combination of utility functions and weights reinitialization scheme to maintain
the network's ability to learn continuous stream of data and tasks.

**Strengths:**

Maintaining plasticity has recently been brought to the attention of the community. Overcoming this problem is key to building an adaptive model in the real world.

It does offer improved performance in some settings when compared with the popular baseline.

**Weaknesses:**

Selecting unused units and reinitializing them has been the baseline method for handling the loss of plasticity problem [1].

The proposed method is a variation of the method in [1] with different utility functions. Instead of reinitializing the unit, individual weights are evaluated and reinitialized.

Similar methods have been proposed in the dynamic sparse training literature.

The finding is purely empirical.

**Questions:**

In most of the tasks and settings, simply using l2 regularization has the best performance. Is more justification needed for the effectiveness of the proposed method?

In [1], the Continuous backpropagation is used with l2 regularization and the combination does offer better performance. In this work, the proposed method is not compared with this combination. What is the performance of the proposed method when compared with this combination?

This work only evaluates the proposed method for supervised learning. Other literature also evaluates their method for reinforcement learning. Compared to supervised learning, reinforcement learning is a more relevant problem as it would be very useful for a deployed agent to keep learning and adapt to new tasks. Is there a reason not to include reinforcement learning as an experiment?

References:

[1] S. Dohare, J. F. Hernandez-Garcia, Q. Lan, P. Rahman, A. R. Mahmood, and R. S. Sutton, “Loss
of plasticity in deep continual learning,” Nature, vol. 632, no. 8026, pp. 768–774, 2024.

---

### Official Review · Reviewer_54p5 · 2024-11-04

**Soundness:** 3
**Presentation:** 3
**Contribution:** 3
**Rating:** 8
**Confidence:** 4

**Summary:**

The paper considers continual learning of neural networks (NNs), and in this context proposes a simple method for maintaining plasticity, i.e., the ability to learn as the problem changes dynamically: *selective weight reinitialization (SWR)*. SWR is proposed to the known method of selective node reinitialization (continual backpropagation). SWR compares favorably, especially on small NNs and modern architectures that employ layer normalization (e.g., transformer). The proposed method is very simple, and has implications for transfer learning as well as continual learning.

**Strengths:**

**Originality:** The paper proposes a simple, yet underexplored idea of reinitializing weights (instead of entire neurons) for the purpose of maintaining NN plasticity. The proposed approach is elegant, and much simpler to implement (and thus arguably more efficient) that the state-of-the-art competitor: continual backpropogation. The authors study different ways in which the weights can be reinitialized, and also consider architectures with and without layer normalization. While the proposed approach does not come first in all the experiments conducted, interesting and original observations are made, e.g., the inferior performance of continual backpropagation in the presence of layer normalization.

**Quality and Clarity:** The paper is very clear, and the methodology is clearly outlined, as well as detailed in the appendices. The proposed method is very simple and straightforward to understand. The paper does not make any ambiguous statements. The paper is practically void of typos and grammatical mistakes.

**Significance:** I believe the implications of the paper are significant, given that (1) transformers are dominating the landscape of modern NN research, (2) continual learning ideas and techniques may be directly transferable to transfer learning, which is widely adopted.

**Weaknesses:**

The authors state that for the transformer architecture, only the feed-forward weights were reinitialized. However, applying the method across all weights, including the attention weights, should be relatively trivial. I feel that "freezing" the attention weights is an unnecessary simplification. I also wonder if the results will be significantly different if attention weights were allowed to reinitialize.

**Questions:**

Page 3, line 142: "We focus our study... to the continual..." -> We focus our study... **on** the continual...

The paper only considers image classification tasks (MNIST, CIFAR datasets). It would have been interesting to see the effectiveness of the method on other data modalities. If not feasible, then perhaps you should clearly outline in the introduction that the paper is focused on computer vision classification tasks.

---

### Note · Authors · 2024-11-25

**Comment:**

We thank the reviewers for the time you’ve invested in providing insightful comments for our paper.

Unfortunately, we’ve decided to withdraw the paper from the conference because we deem it not ready for publication yet. We consider that, as some of you rightfully pointed out, our evaluations don’t properly emphasize the strengths of selective weight reinitialization over continual backpropagation. Moreover, we plan to extend our experiments to reinforcement learning since it’s particularly important for plasticity research.

Based on your comments, we are confident that the next iteration of the paper will be significantly improved. Once again, we appreciate the time and effort put into reviewing our paper.

**Withdrawal Confirmation:**

I have read and agree with the venue's withdrawal policy on behalf of myself and my co-authors.